# DNA-RNA Hybrid (R-Loop): From a Unified Picture of the Mammalian Telomere to the Genome-Wide Profile

**DOI:** 10.3390/cells10061556

**Published:** 2021-06-19

**Authors:** Minoo Rassoulzadegan, Ali Sharifi-Zarchi, Leila Kianmehr

**Affiliations:** 1Université Cote d’Azur (UCA), INSERM-CNRS, IRCAN, 06107 Nice, France; 2Computer Engineering Department, Sharif University of Technology, Tehran 1458889694, Iran; sharifi@sharif.edu; 3Animal Sciences and Biotechnology Department, Faculty of Life Sciences and Biotechnology, Shahid Beheshti University, Tehran 1983963113, Iran; kianmehr96@gmail.com

**Keywords:** sperm, TERRA, telomere, R-loop, genome-wide transcripts

## Abstract

Local three-stranded DNA/RNA hybrid regions of genomes (R-loops) have been detected either by binding of a monoclonal antibody (DRIP assay) or by enzymatic recognition by RNaseH. Such a structure has been postulated for mouse and human telomeres, clearly suggested by the identification of the complementary RNA Telomeric repeat-containing RNA “TERRA”. However, the tremendous disparity in the information obtained with antibody-based technology drove us to investigate a new strategy. Based on the observation that DNA/RNA hybrids in a triplex complex genome co-purify with the double-stranded chromosomal DNA fraction, we developed a direct preparative approach from total protein-free cellular extract without antibody that allows their physical isolation and determination of their RNA nucleotide sequence. We then define in the normal mouse and human sperm genomes the notion of stable DNA associated RNA terminal R-loop complexes, including TERRA molecules synthesized from local promoters of every chromosome. Furthermore, the first strong evidence of all telomeric structures, applied additionally to the whole murine sperm genome compared to the testes, showed reproducible R-loop complexes of the whole genome and suggesting a defined profile in the sperm genome for the next generation.

## 1. Introduction

In the recent period, increased attention has been drawn to R-loop complexes, which appear as frequent features of the eukaryotic genomes. These three-stranded structures are made of the two strands of a DNA molecule, one of them displaced by hybridization with a complementary RNA. Moreover, short-lived intermediates generated during transcription and detrimental effects of stable local R-loops were documented in several instances, while they have also been considered as potentially important players in genome biology [1,2]. However, the multiplicity of possible functions are in contrast with the limited number of analytical methods, which are essentially based on the recognition of the DNA-RNA hybrid either by the monoclonal antibody S9.6, or a catalytic deficient RNase H, coupled with DNA-seq and less frequently, with RNA-seq analysis (reviewed in refs [2,3,4,5,6,7]). Methodological problems are illustrated by the variety of conclusions reached in different studies [2].

One physiological instance of such a three-stranded hybrid structure has been proposed for the mouse and human telomeres [6,7,8,9]. Repetitive double-stranded DNA motifs (TTAGGG)^n^ at the end of all chromosomes and its transcription (UUAGGG)^n^ make telomeres a privileged system to study R-loops [3,4,5]. TERRA, meaning telomeric repeat-containing RNA (TERRA), is an RNA polymerase II (RNAPII) transcript with arrays of G-rich repeats (UUAGGG)^n^ and sequence elements from the subtelomere [4,5,6]. TERRA have also been reported at the telomeric ends of several organisms and near the inactive X in mammals (see review [3]).

In fact, TERRA are thought to be transcribed from a number of chromosomes’ telomeric repeats [6]. These heterogeneous (>100 nt to >1 kb) long non-coding RNAs are found as DNA/RNA hybrids, and a fraction of TERRAs have characteristics of mature mRNAs such as trimethylated cap/polyadenylated at their ends [7]. The origin of the TERRA evidenced on the telomeres is debated. It could be simply maintained after transcription in cis as most of the DNA/RNA R-loop described so far [1,8]. However, a few authors claim a provenance in trans [9]. TERRA detection by Fish assays highlights the end of all chromosomes but did not inevitably reveal the presence of R-loops from which chromosome TERRA originates. Genome-wide mapping by DNA/RNA immunoprecipitation by S9.6 [10,11] antibody followed by high-throughput DNA sequencing (DRIP-seq) of R-loops in human, mouse, and yeast cells have shown the existence of telomeric R-loops but were problematic to definitely establish their origins because of important false signals routinely observed with immunoprecipitation methods and also because DNA fragments are often sequenced and not RNAs.

Here, we directly investigate the evidence of TERRA in germ cells, as sperm cells are transcriptionally silent with limited RNAs left during spermiogenesis to understand how TERRA containing R-loops formed during the last transcription in spermatids are connected to each chromosome and transferred from within sperm head to an oocyte.

Following early reports of a positive DRIP assay [10,11], a consensus seemed established of an R-loop structure involving the terminal DNA repeats of the chromosomes and the complementary TERRA. However, evidence was limited to the telomerase-negative human tumor cells (ALT tumors) without any experimental evidence in a healthy human cell, nor in any mouse cell [9]. We developed an antibody-independent assay for the detection of stable R-loops. Based on the isolation and sequencing of complementary RNAs co-purified with the chromosomal DNA backbone, it can be applied to any biological cell or tissue. This method produces highly reproducible results. First tested on the sperm cells from human and mouse telomeres, it established that in both species, TERRA molecules transcribed from every adjacent sub-telomeric promoters are irrefutably engaged in terminal R-loops. In addition, we integrate the search of enriched peaks from RNA-seq data to discover reproducible non-telomeric TERRA repeats as DNA/RNA regions throughout the mouse genome. These results also highlight and confirm the known pairing region of the two sex chromosomes (X and Y), which are also present as a DNA/RNA hybrid. How the TERRA example might reflect a stable R-loop structure in general is currently unknown. Here, we used similar research methods and applied them to mouse genomes at multiple sites revealing a number of stably DNA-associated RNAs in germ cells sensitive to RNaseH cover the genome. Finally, we demonstrate the robustness of a strategy for the direct in vivo detection of RNA molecules associated with DNA hybrids in highly differentiated germ cells.

## 2. Materials and Methods

### 2.1. Mice

The experiments described herein were carried out in compliance with the relevant institutional and French animal welfare laws, guidelines, and policies. Testes, sperm biopsies, and embryos of *Terc^−/−^* animals in generations G1 to G3 were kindly provided by Dr. C. Gunes (Germany) and Dr. A. Londono (Paris) and *Tert^−/−^* animals JAX mice (Jax strain B6, 129S-terttm1Yjc/J, stock# 005423) were purchased from The Jackson laboratory, Bar Harbor (experimental results of the latter are not shown).

### 2.2. Sperm Preparation

Mouse spermatozoa were collected from the cauda epididymis. Motile spermatozoa were washed twice in MEM buffer (1 mM Na pyruvate, 0.5 mM EDTA, 50 U/mL penicillin, 50 mg/mL streptomycin, and 0.1% BSA) by centrifugation. Sperm pellets were suspended in phosphate-buffered saline (PBS) and centrifuged again. The pellets were washed twice in 50 mM HEPES buffer pH 7.5, 10 mM NaCl, 5 mM Mg acetate, and 25% glycerol. Samples of human sperm of unknown donors were kindly provided by the fertility clinic (20 September 2011 committee number: 2011/10).

### 2.3. Gel Electrophoresis of the DNA/RNA Complex

After overnight incubation at 56 °C in Tris buffer 20 mM pH 8, EDTA 50 mM, with 0.5% SDS, 20 µM dithiothreitol and 400 µg/mL Proteinase K, the total nucleic acid fraction was precipitated with ethanol, re-dissolved, further fractionated on Zymo-SpinTM columns (ZYMO-RESEARCH CORP, Irvine, CA, USA) and digested with Msp1 for resection of the telomeric and subtelomeric sequences up to the first CCGG site. After electrophoresis on either 8% acrylamide or 1.5% agarose gels and transfer to a nitrocellulose membrane, DNA and RNA were revealed by hybridization with ^32^P-labelled oligonucleotide probes.

### 2.4. Preparation of DNA-Bound RNA

Extraction of total nucleic acids after enzymatic removal of the proteins was followed by fractionation of DNA and RNA with the standard TRIzolTM protocol (Rio et al., 2010). Both fraction of nucleic acid aqueous phase (RNA) and chloroform-water interphase separately ethanol precipitated, again, once more fractionated through binding onto Zymo-SpinTM columns (ZYMO-RESEARCH CORP, Irvine, CA, USA) and elution columns or by chloroform extraction (www.zymoresearch.com accessed on 2 March 2021, ZR-Duet ™ DNA/RNA MiniPrep Plus catalog number D7003). The free RNA was separated, and the fraction that remains bound to the DNA further purified after DNase digestion and purification with Zymo-SpinTM columns (ZYMO-RESEARCH CORP, Irvine, CA, USA).

### 2.5. Enzymatic Assays

Restriction cleavage: extracts were incubated overnight with the indicated restriction enzymes (Promega) in 50 µL containing 1–5 µg total DNA. For electrophoresis, 25 µL samples were loaded per lane. For extensive hydrolysis, extracts (20 µL) were incubated for 30 to 60 min with RNase 0.5 µg/µL (which remove all RNA molecules), RNaseH 1 u/µL (which remove only RNA molecules hybridized with the DNA), and DNase 10 u/µL (which remove all DNA fragments), all provided by Roche Life Science (DNase free RNase ref. 11119915001, RNaseA ref. 10109169001, RNase H ref. M0297S).

### 2.6. Sequence Establishment

DNA-bound RNAs were prepared from epididymal sperm 10 to 20 × 10^6^ spermatozoa/male and from the total testis (estimated 60% of diploid cells) of a 6 month-old and a 14 month-old male, and two sperm samples from a man at reproductive age, all showing identical electrophoretic profiles of the DNA/RNA complexes. Mouse spermatozoa were collected from the cauda epididymis and ejaculated human sperm kindly provided by the Fertility Clinics of the University. In both cases, the standard protocol was already reported previously by Kianmehr et al. [12], briefly: motile spermatozoa were recovered and washed MEM buffer (1 mM Na pyruvate, 0.5 mM EDTA, 50 U/mL penicillin, 50 mg/mL streptomycin, and 0.1% BSA) by centrifugation at 3000 rpm. Sperm pellets were suspended in phosphate-buffered saline (PBS) and centrifuged again. The pellets were washed twice in 50 mM HEPES buffer pH 7.5, 10 mM NaCl, 5 mM Mg acetate, and 25% glycerol. Samples of human sperm were identically processed. TRIzol extracted interface materials were ethanol precipitated and followed by overnight incubation at 56 °C in Tris buffer 20 mM pH8, EDTA 50 mM, with 0.5% SDS, 20 µM dithiothreitol and 400 µg/mL Proteinase K. Total nucleic acids after enzymatic removal of the proteins was followed by fractionation of RNA and DNA-bound RNA. Extracts were fractionated by the ZR-Duet ™ DNA/RNA MiniPrep Plus catalog number D7003 protocol according to the specifications of the manufacturer (www.zymoresearch.com accessed on 2 March 2021) see Appendix A. In all cases, the fraction that remains bound to the DNA was further purified after DNase digestion.

After DNase digestion, 10–100 ng of RNA were recovered and sent to Eurofins Genomics (Eurofins Medigenomix GmbH, Ebersberg, Germany) for high throughput sequencing on Illumina HiSeq 2500 or Illumina MiSeq. About 10–100 ng of RNA were analyzed by high throughput sequencing on Illumina HiSeq 2500 or Illumina MiSeq (Eurofins Medigenomix GmbH, Ebersberg, Germany). Libraries were generated and designated D1 and D3 from the DRNAs of a 6 month (D1) and a 14 month-old male (D3), with R1 and R3 the corresponding nucleoplasmic sperm RNAs. Totally, 66 million (m) reads were mapped to unique sites in the mouse genome (D1), 69 m (D3), 77 m (R1), and 87 m (R3). From the total testes it was prepared only the DRNA libraries TD1 (70 m reads) and TD3 (82 m), and from the human sperm of a healthy unknown donor, the hD 1 and 4, 50 m (reads), and hR 1 and 4 (88 m). The average length of the reads was about 100 nt. All the primary sequence characteristics for sample libraries of the DNA-bound RNA and free-RNA sequences of germ cells are summarized in Appendix A. Sperm and testes transcripts profiling and alignment-based visualization were performed for analysis. Comparison of transcript abundance between samples as measured by RNA-seq yielded TPM (transcript per million) values for the different samples.

### 2.7. Sequence Analysis

With FastQC version 0.11.5 (http://www.bioinformatics.babraham.ac.uk/projects/fastqc/ accessed on 2 March 2021) quality control of the RAW sequencing reads was performed. Trimming of bad-quality reads was performed using Trimmomatic version 0.36 [13]. Based on the FastQC results, 10 nucleotides were cropped from the 5′ end of each read, and trimmed bases with Phred score lower than 20 from the leading and trailing ends of each read. The Illumina adapter sequences were also removed, and the trimmed reads with a size less than 30 bp were removed. Hisat2 was used to align trimmed reads to the reference assembly (GRCh38 for human and GRCm38 for mouse) [13], with quality, alignment, and plot-bamstats utilities of samtools [14]. HOMER version 4.9 was used for peak finding and generating bedGraph files for visualization of alignment [15]. The bedGraphs were cross-sample normalized to have 10^7^ reads per sample in order to make visualization of different samples comparable. Visualization of alignment was performed using IGV version 2.3.67 [16]. Genome-wide locations of TERRA repeats were found by aligning sequences of length 24 and 48 composed of 4 and 8 copies of TERRA to the reference genome using bowtie2 using “-a” argument to report all alignment [17]. To direct quantification of reads, reads with a length of less than 20 bp were discarded. Samples were then directly quantified using Salmon v0.12.0 [18] on an index built from the GRCm38 genome using GENCODE vM18. RNA composition estimation were created by summing the reads of transcripts from the same RNA biotype and dividing by the total number of assigned reads in each sample [19]. The average percent of TERRA (minimum four TAACCC repeats) in raw sequencing reads (Fastq files) counted using ‘grep –c’ command. We also detected statistically significant peaks of expression using HOMER (4-fold expression over local regions, Poisson *p*-value over local regions <0.0001, and FDR adjusted *p*-value < 0.001). We retrieved the sequences of those peaks located in telomeric regions using UCSC genome browser. Multiple sequence alignment of these peaks was performed using EMBL-EBI Clustal Omega [15] and the percent identity of the alignment on matrix was visualized with the heatmap R package.

## 3. Results

### 3.1. Antibody-Independent Profiling of R-Loop Complexes: A General Approach

We devised a simplified, quick, and inexpensive method for the detection of DNA/RNA hybrid regions in a complex genome (Appendix A). We first validate with telomeres associated TERRA, taking advantage of its abundance as a repetitive element with multiples sites of detection. Unlike the current DRIP assay, this is a direct approach, which is not based on antigenic recognition or RNase H recognition of the hybrid, with the advantages of avoiding possible artifacts. A variant method was developed to obtain stable DNA/RNA hybrids that are maintained during the TRIzol chloroform extraction procedure [20,21]. It was established that the TRIzol reagent permits the sequential recovery of RNA, DNA, and proteins. First, cells were lysed with TRIzol buffer, adding chloroform led to separation of water and phenol/chloroform phases, a clear upper aqueous layer with RNAs, interphase, and a red lower layer with DNA and protein. RNAs were recovered by precipitation with isopropanol from the aqueous phase while DNA was recovered with ethanol precipitation from the interphase. In the most generally used TRIzol chloroform extraction procedure [20,21], a small but reproducible amount of RNA is retained, together with high molecular weight genomic DNA, at the chloroform-water interface, while the free RNAs molecules of a variety of molecular sizes are found in the upper aqueous phase. The material recovered from the chloroform-water interface as a whole (genomic DNA with DNA/RNA hybrid) was dissolved in Tris-HCl 10 mM, EDTA 1 mM, and immediately ethanol precipitated to remove trace of Trizol and chloroform. The pellet was then dissolved in proteinase K hydrolysis buffer to completely remove all proteins. DNA and DNA/RNA complexes were then recovered by chromatography on a DNA binding Zymo-SpinTM column (Zymo-Research Corp Irvine CA, USA) to eliminate all forms of remaining impurities present in the extraction, free RNAs, proteins, and degraded nucleic acids. As illustrated below in the case of the telomeric complex, a specified genomic area could be analyzed separately by cleavage of genomic DNA with an appropriate restriction enzyme followed by gel electrophoresis, in the absence of a nucleic acid denaturation step and direct transfer from the gel to a nitrocellulose filter. Negatively charged nucleic acids (single strand DNA and RNAs) are efficiently retained on the positive charged nitrocellulose membrane, while double stranded DNAs fragments are poorly or not at all retained. These conditions have been previously documented in T. Maniatis’ classic molecular biology book from (Molecular cloning: A laboratory Manual) which allow the transfer of single-stranded RNA or DNA fragments while double-stranded fragments are not retained on the membrane. In addition, after transfer to a nitrocellulose membrane, the signals are revealed with a specific validated probe of telomere with motif (CCCTAA) [8] repeats to reveal repetitive motifs of telomeres containing Leading (DNA) and TERRA fragments. Alternatively, whole preparations of the DNA-associated RNA were obtained by extensive hydrolysis of pancreatic DNase and a further RNA purification step via the Zymo-SpinTM RNA binding column (Zymo-Research Corp Irvine CA, USA) leads to DNA associated fraction of the RNAs which here we call the D-RNAs fraction for RNA sequencing analysis compared to the R-RNAs of the free fraction (see Appendix A).

### 3.2. The Telomeric TERRA Complex

The ends of the human and mouse linear chromosomal DNA molecules of individual chromosomes are long TTAGGG/CCCTAA repeats. After the discovery of the homologous UUAGGG-repeated TERRA [4,5,6,7,8,9,10,11,20,21,22], the DNA telomeric repeats were thought to be engaged in three-stranded R-loops with TERRA. Attempts in normal tissues to directly show evidence of R-loop structures has so far only been partially successful. In cell culture, mostly from two human pathologies, could R-loops be revealed by DRIP assay, namely telomerase-negative (ALT) cancers [23] and ICF syndrome cells [24]. The structure of the telomeres of healthy human cells, as well as of any murine cells, remains to be directly shown. One main site of TERRA transcription was identified in mouse and human cells [9,25], and it has even been considered that transcription from individual sub-telomeric promoters on every chromosome is a unique characteristic of a class of malignancies (ALT cancers).

Here, we generated results in parallel on human and mouse sperm cells. Sperm cells were chosen for the first trial because they have silent transcription, and the majority of cytoplasmic RNAs are already removed at the compacting stage of spermiogenesis, thereby minimizing the risk of contamination by cytoplasmic RNAs. The total nucleic acid fraction recovered from the TRIzol chloroform-water interface was ethanol precipitated and after purification (see Appendix A and Methods for protocol) was subjected to Msp1 (CCGG) restriction enzyme cleavage for resection of the DNA telomeric sequences up to the first upstream CCGG site, less than an average of 2000 bp from the repeats on every chromosome. Each sample was then submitted to gel electrophoresis in 8% acrylamide which produced a sharper band followed by electro-transfer to nitrocellulose under conditions of binding of the single-stranded RNA and DNA material. Without the NaOH step before the transfer, double-stranded nucleic acid materials in this condition are not retained on the nitrocellulose membrane, and only hybrid nucleic acid fragments with single-stranded nucleic acid (DNA or RNA) or hybrid molecules containing a region of “single-stranded” DNA or RNA molecules with double-strand fragments such as telomeric R-loops structure are retained. The positively charged nitrocellulose membrane retains single-stranded RNA or DNA. After hybridization with oligonucleotide probes complementary to the telomeric repeats and TERRA revealed a strong, unique homogeneous signal. Identical results were generated in every cell type tested, exemplified in Figure 1, for three laboratory healthy mouse lines (*C57BL/6*, *B6D2*, and *129/sv*) and the three tissues analyzed, brain (B), total testis (T), and purified epididymal sperm (S). Figure 1a shows DNA/RNA complex detection, the pattern of the material from the chloroform-water interface DNA-associated RNA fraction that we named “Tatar-blot” for Telomere associated TERRA. The probe (CCCTAA)^n^ used in Figure 1a reveals Leading strand (5′-3′) telomeric DNA and, if present UUAGGG TERRA (see Figure 2 for evidence). In addition, the presence of single-stranded DNA or RNA or DNA/RNA hybrid fragments in the complex is the only condition that will allow the DNA fragments to be maintained on the nitrocellulose membrane, especially without a double-stranded fragment denaturation step. With the same conditions and protocol applied to human sperm, we also observed one major signal. Visualization of the complex requires resection from chromosomal sequences. The clearest results were generated by cleavage with Msp1 restriction enzyme at (CCGG) sites present at least once in the immediate subtelomeric region less than 1–2 Kb from the ends of all mouse and human chromosomes so that the contribution of chromosomal sequences is minimal to facilitate transfer and retention on the nitrocellulose membrane. BamH1, Bgl2, Alu1, Mse1, and Dpn1 restriction endonucleases tested (Figure 1b) did not generate comparable results, most probably because they cleave at greater and variable distances from the telomeres, thus generating larger double-stranded fragments, a limiting factor for efficient retention on nitrocellulose. Figure 1b shows BamH1, Msp1, HpaII and Alu1 restriction enzymes with nucleic acid materials collected from mouse tissues (brain, testis, and sperm) and human saliva. In agreement with previous observations of increased methylation of CpG sites in subtelomeric regions [26], cleavage with Hpa2, a methylation-sensitive isoschizomer of Msp1, failed to show evidence of the complex (Figure 1b). Specificity was determined by hybridization of the transfers to radioactive probes for other repetitive (SINE, Xist-A) or single-copy genomic sequences (sequences can be seen in Appendix A), none of them generating a significant signal. Electrophoretic migration of the complexes was always equivalent to that of a high molecular weight linear DNA molecule (Figure 1a,b) but was clearly not informative as to the actual size. Here, with a universal probe that reveals all telomeric repeats, the sizes of the fragments were not studied due to the complex structure of R-loop with variable repeat length and also because R-loops migration under these conditions does not sufficiently allow for separation of the fragments to visualize individual chromosomes. In addition, the absence of NaOH and HCl treatment in “TaTar-blot” analysis is the only way to preserve the structure of the R-loop of the DNA/RNA hybrid intact. It is why the size determination of the fragments was not informative at this level of analysis.

When the same extracts were further processed for Southern blot analysis on agarose gels (0.8%) followed by 45 min in 0.5 N sodium hydroxide (a condition that denatures double-stranded DNA and removes all RNA), transfer and hybridization (Figure 1b) revealed signals with high molecular weight molecules and below with a lower signal smeary profile (right with longer exposure of autoradiogram) starting with the expected large size of the mouse telomere (>12 kb). It is known that mouse telomeres are long with variable lengths and that to separate these long DNA fragments requires pulsed-field gel electrophoresis; therefore, routine agarose electrophoresis gel does not separate very long DNA fragments. To reveal large fragments of the DNA after pulsed-field gel electrophoresis NaOH and HCl treatment steps of the agarose gel are required before transfer to the membrane. NaOH and HCl treatment destroys the R-loop hybrids, which is why this could not be used in “Tatar-blot” protocol.

In addition, these complexes did not appear to depend on the telomerase, as they were observed in three generations of telomerase-negative mouse sperm cells *Terc^−/−^* (Figure 1c). The same profile was generated by the probe CCCTAAn (Figure 1c top), which revealed TERRA and a single-stranded leading telomeric DNA strand, and the TTAGGGn probe (Figure 1c bottom), which revealed a lagging DNA strand. These results revealed that complexes transferred were bound to the membrane with both single-strand telomeric DNA and RNA as well. The overall signals revealed between the first and third generation of *Terc^−/−^* did not vary much, possibly because the telomere shortening timing in sperm cells is different from reported somatic cells, and this last point requires further investigation. The same results were also obtained with *Tert^−/−^* sperm (not shown). In the case of the *Tert^−/−^* and *Terc^−/−^* mutants, they could also be evidenced in embryonic fibroblast cultures from homozygous crosses during the first three generations in which the mutants maintain telomeres [27,28] and after immortalization in a culture of mutant cell lines (M.R., unpublished).

Complexes with the same electrophoretic profiles were identified in different somatic cell types. To minimize contamination of the cytoplasmic RNA, TRIzol extraction was applied twice on the somatic cells. DNA-associated RNAs are stably maintained with genomic DNA even after twice treatment with TRIzol. Comparable profile and amounts in every mouse cell tested, from embryonic fibroblasts to adult tissues (testis, brain, liver, and kidney) as well as cultivated cell lines and short-term cultures of differentiated cells, in Figure 1a,b are exemplified for mouse brain and testis extracts. This was also the case for human tissues (Appendix A), including saliva, blood, and sperm (Figure 1a,b). Identical electrophoresis profiles were generated for murine and human sperm extracts by probes against the CCCTAA motive (Figure 1a). To evaluate nucleoplasmic free TERRA molecules, RNAs were precipitated with isopropanol from the water phase of the TRIzol extraction and analyzed in Figure 1d by Northern blot assay (1% denaturing agarose gel with formaldehyde), the oligo probe (CCCTAA motif) revealed the 3 kb major TERRA fragments recovered from the TRIzol aqueous fraction. The quantity of the free TERRA/cell number (10^6^ cells/lane) extract was much higher in the testes and brains compared to sperm cells.

Properties of the complexes were compatible with the R-loop structure schematically drawn in Figure 2 (Figure 2a), as revealed by the G-rich strand of telomeric DNA displaced by the TERRA hybrid. To evaluate nucleic acid complexes in Tatar assay (Figure 2b), nucleolytic attack was performed by incubating extracts (20 µL) for 30 to 60 min with either RNaseA 0.5 µg/µL (which removed only free strands of RNA and not RNA hybridized with DNA), RNaseH 1 u/µL (which removed only RNA hybridized with DNA), or DNase 10 u/µL (Figure 2b) which removed DNA. The electrophoretic signal disappeared after incubation with DNase (removal of all DNA), pancreatic RNase (removal of all RNA), formaldehyde (completely denatured R-loop complexes), or RNaseH, while it was not affected by RNase A (Figure 2b). Higher signal with RNase A could be explained by the removal of the free strand of RNA not hybridized with DNA that could facilitate migration and transfer of the complexes to the membrane. Another feature of the complex was the slower rate of migration during gel electrophoresis performed in the presence of 20 µM PhenDC3(4), which reacts with G quadruplexes [29] that blocked the structure (Figure 2b) as the band was shifted to the top of the gel.

The same results were also generated in parallel on a series of human and mouse cells, ex vivo organ extracts that were mostly normal cells, but included cultured cell lines (Appendix A), and the human U-2 OS cancer cell line (ATCC^®^ HTB-96™) as a positive control, in which DNA/RNA telomeric hybrids have previously been identified by DRIP [23].

### 3.3. Transcription of Individual Telomeres from Local Subtelomeric Promoters

A key aim of this paper was to establish whether the observed structure was present in only one or a few telomeres or, on the contrary, was a general feature of the genome. It has been generally assumed that R-loops are generated by transcription from adjacent promoters. On the other hand, a major part of TERRA has been reported in cell culture to be transcribed from chromosome 18 in mouse cells and from chromosome 20q in humans [9,25] so that our data are so far compatible either with a unique R-loop on these chromosomes or with binding in trans to other chromosomes of RNAs from the mouse 18 and human 20q loci. However, it appeared that unique subtelomeric sequences of different chromosomes are represented in the RNA from the DNA-associated fraction. We generated probes for sequences between the first 5′ Msp1 site just before the G-rich repeats of the telomeres of chromosomes 9, 17, 18, 19, X, and Y telomeres and probes for the complementary strand up to the C-rich repeats (indicated “xxx/x’x’x’” in Figure 2a). By using the UCSC genome browser Blat tool [30], we ensured that these flanking regions are unique in the whole genome. All the probes (Appendix A) generated positive hybridization signals in the complex revealed by “Tatar blot” as illustrated in Figure 3 for the sub-telomeric sequences of chromosomes 9, 17, 18, and Y. As controls chromosomal S6 (a repetitive sequence) and none of the probes for sequences on the centromere side of the terminal Msp1 restriction enzyme sites generated signals. “Tatar blot” analysis indicates specific nascent TERRA from each chromosome ends.

We further ascertained by Illumina high-throughput analysis of the DNA-bound RNA molecules (see Appendix A for RNA preparation) that their sequences are compatible with transcription in each telomere from a local promoter Figure 4 (Appendix A). Starting from purified mouse sperm, we first confirmed by RNA-seq that a fraction of the DNA-associated RNA molecules amounting to 0.1 percent of the reads showed the characteristic UUAGGG TERRA repeats (minimum four repeats, Appendix A). We then searched the RNA sequence libraries for chromosomal sequences next to the repeats and extended the search in the 5′ direction to collect upstream flanking regions in Figure 4a. We again ensured that these flanking regions are unique in the whole genome. As exemplified for chromosomes 2 and 10 in Figure 4b, their 5′ extension in the mouse genome indicated that nascent TERRA originates in each case from promoters in the immediate sub-telomeric region. Complementary evidence for the synthesis of these RNAs from distinct promoters is shown in Figure 4b, with the heat map showing the percent identity matrix between each pair of sequences of chromosomes 2 and 10. The maximum identity is 93%, thus making it impossible that these reads could originate from a unique chromosome (Figure 4b). The available sequence data and libraries allowed us to reach the same conclusion for chromosomes 2, 3, 5, 10, 12, 13, 14, 16, 19, and X see Appendix A. R-loop structures involving the products of local transcription, therefore, appear as a general feature—especially taking into account some constraints in the analysis, for instance, the fact that in the latest mouse assembly (mm10), chromosomes 4, 6, X, and Y remained to be sequenced up to the repetitive telomeric tract. Testes samples TD1 and TD3 also contain nascent TERRA in their DRNA fractions, as shown in Appendix A. In conclusion, Illumina high-throughput analysis of the DNA-bound RNA molecules completely confirmed the results of the molecular analysis of the “Tatar blot”.

DNA associated TERRA found from every telomeric end in mouse and human sperm cells is consistent with the idea that TERRA is expressed from subtelomeric promoters and remained associated at the telomeres ends. TERRAs that remained bound to the sperm genome were generated from the last wave of transcription before the compaction step at the late stage of spermiogenesis. Our results need to be compared with previous reports of major TERRA promoters in C18 (mouse) and 20q (human) chromosomes [9,25]. There is, in fact, an apparent contradiction since these promoters were identified in experiments in which TERRA was prepared by standard TRIzol extraction from abnormal cells such as cancer cells, and thus correspond to the nucleoplasmic RNA fraction (free RNA fraction in aqueous phase) thought to be involved in other, non-telomeric functions [31,32]. Thus, it is important to recover TERRA from both nucleoplasmic and DNA-associated fractions.

### 3.4. Extension Analysis to Other DRNAs with UUAGGG Motif Repetition

Since the association of TERRA transcripts with telomeres could be revealed by the proposed method, we also asked if TERRA hybrids located at defined non-telomeric sites could be detected from data generated by Illumina high-throughput analysis of the DNA-bound RNA molecules. In order to try to distinguish stable complexes of R-loops [8,33] and the transient association of the nascent RNAs during the process of transcription, we attempted to verify the data from the mouse DRNA-seq (mouse sperm and testes). In Appendix A for a general evaluation of the RNA molecules associated with DNA (indicated “DRNAs”), the RNA-seq analysis was extended to all genomic regions containing transcripts in mouse sperm (indicated D1 and D3) and in the total testicular tissues, (TD1 and TD3) in the Appendix A.

Genome-wide profiling shows representative landscapes of expression signals of chromosomes 2, 10, 12, and 18 (see Appendix A). For each chromosome, the green bar shows the location of TERRA-homologous regions (minimum four consecutive TTAGGG repeats, necessary to form respective G-quadruplex structures), which are present at several chromosomal locations in DRNA fractions.

In addition, research on known non-telomeric TERRA sites has identified a fraction of TERRA transcripts near the sex chromosome that undergo pairing in both sexes, the inactive X chromosome (Xi) of female cells, and the Y [32] chromosome (see Figure 5a). The Asmt locus on the X chromosome and Erdr1 locus on the Y chromosome have been proposed to pair together via UUAGGG repeats. An example of transcripts is visualized (Figure 5a) within the pairing center in the testes and sperm RNA data. As shown in Figure 5a, DRNAs transcripts from the Erdr1 locus with UUAGGG repeats were present on the Y chromosome. On the other hand, the Asmt transcripts were present on the X chromosome. These results indicate retention of PAR-TERRA transcripts in pseudoautosomal regions of X (Asmt) and Y (Erdr1) in the DNA fraction of the mouse sperm and testes. Other regions are exemplified with non-telomeric TERRA sites in Figure 5b, on chromosomes 2, 9, 14, and Y, respectively.

### 3.5. DRNA Is Not Restricted to the UUAGGG Repeats

Illumina high-throughput RNA-seq analysis of the DNA-bound RNA molecules of sperm and testicles of mice revealed transcripts from the whole reference genome. Here we only analyzed data from mice because DRNA-seq data was available for two types of samples: sperm (haploid) and testis containing both haploid and diploid cells (meiotic, diploid, and haploid germ cells and diploid somatic cells). A general overview of RNA associated with DNA revealed, in addition to TERRA, a wide variety of RNA sequences maintained on the DNA (DRNA fraction), as shown for mouse sperm and testes in Appendix A. In the sperm DRNA fraction, only a few significant peaks were observed at several sites in the genome and were also present in all samples (R and TD) with high reproducibility, while a number of low signals were present along all chromosomes, specific to the types of preparation (homolog between D or homolog between the R fractions). To further characterize these relevant peaks, we performed a search for sequence homology. These sequences were transcribed from different genomic regions see Appendix A with an average of 1 kb and did not contain sequences or homologous motifs. To further test the possibilities offered by the analysis, we compared the distribution between the DRNA of the main hybrid regions and the unbound rRNA between sperm and somatic (testis) cells. The annotated positions of the promoter-TSS (transcription start site), TTS (transcription termination site), exons, introns, and other characteristics were based on mm10 transcripts. In Figure 6, a typical peak annotation bar plot shows that most of the peaks in sperm DRNA samples fell into enhancer regions (intergenic and intronic regions) compared to all others. While most of the peaks in sperm-free RNA and testes DRNA fell in the exon regions. In the intergenic region, mostly simple repetitive sequences were observed (Appendix A). In Appendix A, the enriched transcripts in different regions are listed. These results revealed that in addition to TERRA, other transcripts were associated with the structure of the R-loop hybrids and were stably maintained on the DNA. Preferential intergenic transcript retention in the sperm DRNA fractions may present functional importance but remain unknown, and their possible existence should not be ignored in future research.

## 4. Discussion

Here, we have documented the following results: (i) a fraction of nascent TERRA was stored packed in vivo with the genome in the form of DNA/RNA hybrids in the telomeres of each chromosome in sperm (mouse and human) and other mouse organs (testes and brain); (ii) non-telomeric TERRA repeats transcribed from different regions of the genome were also maintained in their genomic loci as DNA/RNA hybrids; (iii) a fraction of the TERRA repeat-containing transcripts previously described near the sex chromosome were attached as DRNA to Y (Erdr1) or X (Asmt) chromosomes; (iv) other RNAs than the TERRA repeats were found to be attached to DNA fractions with simple repeated sequences; (v) a general profile of transcripts stably attached to DNA throughout the mouse genome and the discovery distinct transcripts profile retained in the sperm DRNA fraction and important differences in the levels amongst the transcripts attached to DNA in sperm and testicular cells. Our results suggest that RNAs associated with DNA could define cell fate and specificities. The approach developed and validated here could be easily applied to any type of cells in normal development and diseases to eventually establish its dynamics.

DNA-associated RNA (DRNA) extraction is a straightforward approach without the need for supplementary interacting guide molecules (such as antibody or gene product), inexpensive, highly reproducible, and efficient. Bioinformatic analysis with alignment to the reference genome allowed the generation of genome-wide profiling of stable DNA/RNA hybrid transcripts purified with genomic DNA. Here, our data shows that all of these stable R-loops provide insight into the amount of RNA maintained in the sperm genome. These transcripts are of interest in view of the various reports highlighting a role for sperm RNAs in the epigenetic control of gene expression [34,35]. We propose that hybrid DNA/RNA interaction in semen could serve to preserve transgenerational RNA signaling, which needs to be taken into account and experimentally tested in functional tests.

The parallel is possible, with S9.6, RNase H, and more recently MapR [36] establishing that GC content is important for the recognition of the R-loop by antibodies and, in particular, by RNaseH [37]. Alternatively, here unlike the previous methods, nascent R-loop detection without protein associated with DNA/RNA hybrids in normal cells (sperm and testes) reveals the stable structure of DNA/RNA hybrids, which were not particularly rich in CpG. The two types of interaction were not mutually exclusive. However, additional factors must be considered to retain these signals. Here, our study shows that the difference between antibody-directed (complex) and direct detection of DNA/RNA hybrids is that, although both appear to detect R-loops, the latter was highly specific for direct DNA and RNA molecule interactions and did not involve any other supplementary configuration or mechanisms.

In the testes, several studies have already reported genome-wide transcripts, and it is also assumed that from the last stage of elongated spermatids, with the elimination of the majority of cytoplasmic RNAs, a number of transcripts remain in the sperm [38]. We have recently reported profiles of DRNA-seq mouse sperm transcripts from all coding and non-coding regions of the genome [12]. Unlike in the analysis of previous individual transcripts [12], here, we performed peak finding in order to associate the location of the peaks and to compare maps in terms of their genomic characteristics.

### 4.1. Genomic Profiles

Genome-wide profiling detected DNA/RNA hybrids at multiple loci across the genome [39] (also see the previous report cited). However, R-loop mapping in the genome is debated in the literature because of differences in mapping findings reported from one report to another. The differences between reports are perhaps mediated by the preference of antibodies for a certain type of structure or sequences that could be varied from one cell type preparation to another. Our current data from the direct detection of the nascent transcripts provide clear, reproducible evidence of stable regions of R-loops that could easily be isolated from any cells.

The bioinformatic analysis of sperm (DRNAs fraction) compared to free RNAs and the testis DRNA fraction revealed a significantly higher level of intergenic transcripts than found in exon, intron, 3′, 5′, promoter-TSS (transcription start site), TTS (transcription termination site), and non-coding transcripts (Figure 6). Some expression signals were enriched in D fractions containing simple repetitive sequences located at intergenic regions. The DNA/RNA hybrid regions detected here consisted mainly of short, simple repetitive sequences. Other repeated elements such as LIN, SIN LTR, and satellites are also present but less frequently (Appendix A). Intergenic transcripts have different characteristics from mRNA and are often involved in chromatin remodeling, transcription control, including enhancer or silencing activities. Therefore, a difference in the level of DNA/RNA hybrids (R-loops) in different regions, in particular a higher level in the intergenic region compared to other regions, could define the specific state of spermatozoa. Although the accumulation of R-loops is associated with genomic instability, here, the DNA/RNA hybrids were detected at the physiological level could have a role in cellular characteristics, serving as a signal for keeping cells in an active or inactive stage. In human spermiogenesis, a large abundance of intergenic RNAs has been reported [38] but to clarify their roles, more experiments are needed to understand how they are involved in a given function. In addition, a recent study suggests that R-loops are involved in functions such as determining cell fate [39].

### 4.2. Genomic Profiles of the TERRA (Telomeric and Non-Telomeric)

In parallel, both molecular analysis and Illumina high-throughput analysis of the DNA-bound RNA molecules (DRNA-seq) was applied firstly to TERRA complexes of the sperm from human and, in particular, mouse sperm telomeres, provide significant solid information. In fact, in both species, sperm telomeres retained a fraction of the TERRA synthesized directly from promoters adjacent to all chromosomal ends. In addition, this was confirmed by molecular analysis in human somatic cells and in a variety of mouse cells in culture and tissues. With DRIP-seq analysis, such a conclusion could not be completely reached. Here, with both molecular and RNA-sequencing analysis, we reach the same conclusion that DNA/RNA hybrids at the level of telomeres were maintained in all types of cells. In addition, RNA-seq analysis in the testes and sperm of mice confirmed that the TERRA (telomeres) transcribed in the testes of all chromosomes were maintained until the final stage of sperm maturation in the epididymis. Human and mouse sperm TERRA would be thus transferred to the offspring packed with genomic DNA. TERRA as DNA/RNA hybrids in the genome suggests a potential role of these non-coding RNAs on the stability of the telomeres and may be on its expression in the next generation.

We addressed that TERRA signals are significantly enriched at telomere sequences; and TERRA repeats are also reported at non-telomeric regions of chromosome 2, 9, 10, 12, 14, 18, and Y (see Appendix A), where we could observe significant signals of non-telomeric TERRA transcripts in the DRNA fraction. In fact, some non-telomeric TERRA loci overlapped with intergenic regions, such as a non-coding RNA (GM30762) on chromosome 9. Since TERRA signals at intergenic regions were underrepresented, the question arises as to why TERRA signals are low at intergenic regions compared to telomeres. In this case, searching for promoter activity should be investigated in future research.

We also identified sex chromosomes PAR-TERRA (UUAGGG)^n^ containing transcripts in the DNA/RNA hybrids fraction of mouse sperm, which were previously established in ES cells [32]. The finding of PAR-TRERRA transcripts stored in sperm on Y and X chromosomes is intriguing because sperm are blocked for transcription and replication. Among others, one objective of future research will be to decipher the underlying mechanism of PAR-TRERRA transcripts that are attached to DNA in the sperm.

Evidence of DNA/RNA hybrids for both TERRA and its generalization to genome-associated transcripts in semen underscore ongoing questions about how these hybrids might influence genetic and/epigenetic memory. Among several interpretations, one possibility is a trivial phenomenon unique to sperm: the RNA produced by the spermatid at the time of conversion of the chromatin to its final compact structure can remain associated with the locus, perhaps only the last oligonucleotide segment synthesized, giving the artifactual image of a complete RNA taking into account the sequencing procedure. However, DNA/RNA hybrids in a way in active testicular cells were also observed. In addition, a noticeable difference in the transcripts level from different regions between sperm and testes was visible. Moreover, our less advanced analysis of somatic cells also showed a direct hybrid DNA/RNA structure detectable at the telomeres. These studies demonstrate that some transcripts appear strongly maintained in a stable complex DNA/RNA hybrid structure, and further studies should reveal that its meaning and function may be intentional and selective for a given cell.

In conclusion, we believe that the approach designed to identify the telomeric TERRA has wider applications. These results are encouraging and should be extended to different processes and diseases. Future research is needed to reveal specificities, biogenesis, and functions of stably DNA-associated RNAs.

## Figures and Tables

**Figure 1 cells-10-01556-f001:**
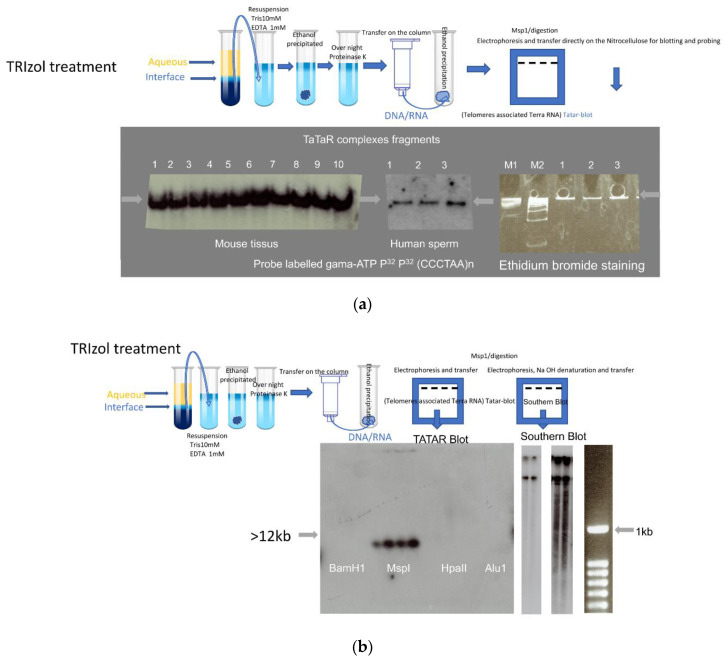
Telomeres associated TERRA. (**a**) DNA-RNA complex (Tatar: Telomeres associated TERRA RNA). Total nucleic acid preparations were analyzed after enzymatic digestion/Msp1. Electrophoresis was run on acrylamide gel (8%) for 3 h and after overnight electro-transfer on nitrocellulose membrane is hybridized with a gamma-ATP ^32^ P-labelled probe against the (TT/UU)AGGG telomeric motif, probe (CCCTAA)^8n^ to reveal Leading DNA strand (telomeres) and RNAs (TERRA). Left: DNA prepared from laboratory mouse tissues *C57BL/6* (1: Brain; 2: testis; 3: sperm), *129/sv* (4: Brain; 5: testis; 6: sperm), *B6/D2* (7: Brain; 8: testis; 9:sperm), and 10: human sperm. Middle: three independent sperm samples. Right: Ethidium bromide staining of markers and three human sperm samples. M1 and M2 molecular markers. (**b**) Left: Enzymatic digestion, BamH1 (1–4), Msp1 (5–8), HpaII (9–12), Alu1 (13–16), (from left to right: Balb/c, B6/D2, C57BL/6 sperm, and human saliva). After enzymatic digestion, samples were run on an acrylamide gel (8%) for 3 h and after were transferred to a nitrocellulose membrane hybridized with an oligo labeled gamma-ATP P^32^ Probe (CCCTAA)^8n^ to reveal Leading DNA strand (telomeres) and RNAs (TERRA). Middle two exposures of the same membrane: Agarose gel (0.8%) classical standard Southern blot, with one step of 45 min 0.5 M NaOH treatment (to remove RNAs and denature double-stranded DNA) before electro-transfer on nitrocellulose membrane and hybridization with the same probe to reveal the (TT/UU) AGGG telomeric motif. In addition to large fragments, a smear of shorter fragments is detected at longer exposure (right). (**c**) DNA/RNA complex with the same protocol as in a and b of mouse sperm from wild type and telomerase-negative (*Terc^−/−^*) mutants of the first three generations (G1, G2, G3). Germ cell biopsies of *Terc^−/−^* animals were kindly provided by Drs C. Gunes (Germany) and A. Londono (Paris). Top: P^32^ Probe (CCCTAA)^8n^ to reveal Leading DNA strands (telomeres) and RNAs (TERRA). Bottom: P^32^ Probe (TTAGGG)^8n^ to reveal Lagging strand DNA (telomeres). (**d**) Northern blot analysis of RNA processed from TRIzol-chloroform aqueous phase, high-level detection of major TERRA fragments in Brain and testes but the low level in sperm (extracted from the same number of cells 10^6^). RNA:B6/D2: 1: Testes; 2: Sperm 3: Brain, C57BL/6: 4: Brain; 5: Testes; 6: Sperm, 129/sv: 7: Testes; 8: Brain; 9: Sperm. P^32^ Probe (CCCTAA)^8n^ used to reveal RNAs (TERRA).

**Figure 2 cells-10-01556-f002:**
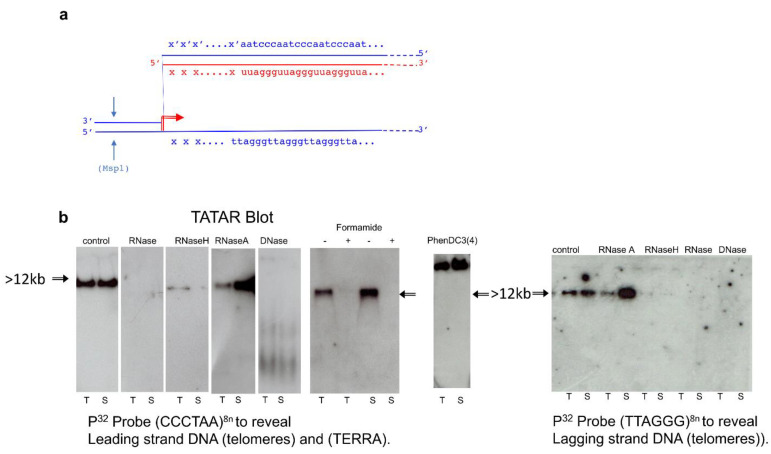
The telomeric TERRA complex (DNA/RNA hybrid). (**a**) A simplified R-loop model. TERRA (red line) is assumed to originate from a subtelomeric promoter (red arrow); blue lines: telomere and sub-telomere DNA; “xxx”: 5′ TERRA sequence from subtelomeric DNA (xxx/x’x’x’); arrows: Msp1 cleavage. (**b**) Telomeres associated TERRA RNA detection after Msp1 cleavage performed as in Figure 1 on the indicated mouse cells (T: testis and S: sperm). Left to right: The complex shown in several conditions: control detection of the (TT/UU)AGGG containing fragments, in extracts treated before loading with -pancreatic (DNase free RNase ref. 11119915001), or-RNase 0.5 µg/µL (which remove all RNA), or-RNaseH 1 u/µL (ref. M0297S) to remove all RNA hybridized with DNA, or-RNaseA 1 u/µL (ref. 101091690 01) to remove single strand free RNA and/or DNase 10 u/µL, to remove all DNA respectively (all provided by Roche Life Science), and control electrophoresis after addition of either 50% formamide (denaturation conditions +/ or −) which denature all nucleic and ribonucleic acid or −20 µM phenylDC(3)4 (right) which blocks quadruplex. Nitrocellulose membrane filters were hybridized with two probes to reveal: at left P^32^ Probe (CCCTAA)^8n^ Leading strand DNA (telomeres), (TERRA) and at right P^32^ Probe (TTAGGG)^8n^ Lagging strand DNA (telomeres).

**Figure 3 cells-10-01556-f003:**
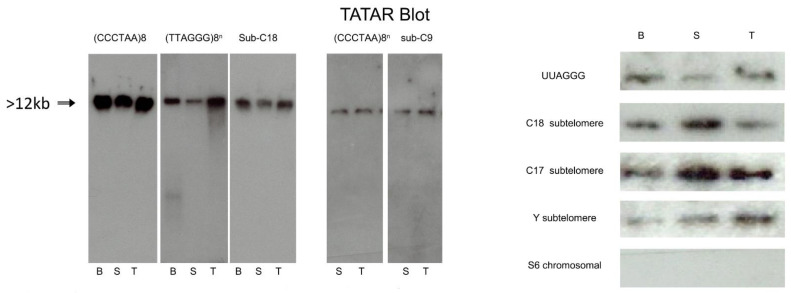
TERRA transcripts originate from subtelomeric promoters. Tatar blot (telomere-associated TERRA RNA) was performed with brain, sperm, and testis extracted DNA/RNA complexes, purified and digestion with MsP1, electrophoresis, transfer of complexes, and hybridization with different probes. From left to right: In parallel membranes are hybridized with P^32^ Probe (CCCTAA)^8n^ to reveal Leading DNA strand (telomeres) and RNAs (TERRA), Probe (TTAGGG)^8n^ to reveal Lagging DNA strand (complementary strand telomeres), Sub-C18 (subtelomeric chromosome 18), and sub-C9 (subtelomeric chromosome 9). Right: different probes (top to bottom) for TTAGGG repeat (eight-fold), subtelomeric sequences of chromosomes 18, 17, and Y (schematized as xxx in Figure 2a), and S6 ribosomal RNA as a negative control (see Appendix A for corresponding oligos nucleotides sequences). The chromosomal sequences were randomly taken from the mm10 mouse genome assembly and checked in each case to be unique in the whole genome.

**Figure 4 cells-10-01556-f004:**
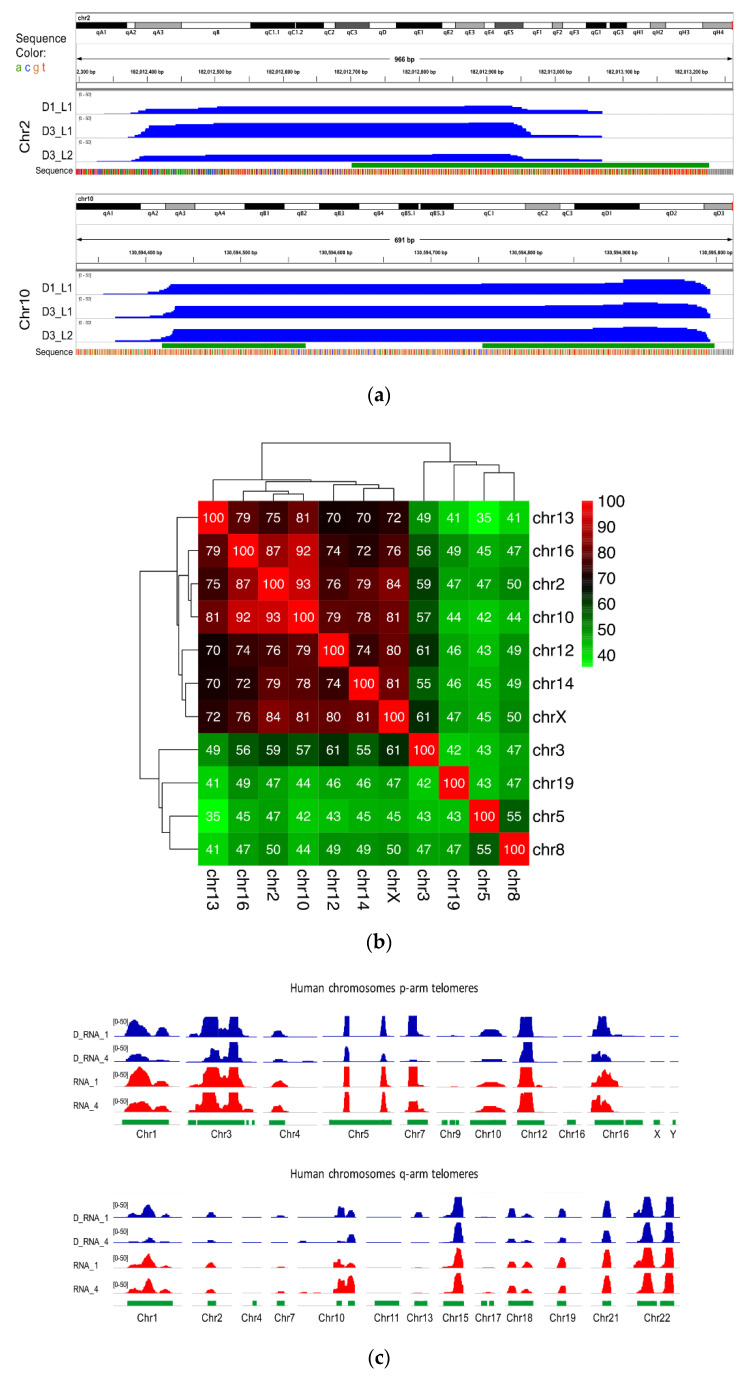
DNA-bound RNA samples over telomeric regions. (**a**) Expression patterns of mouse DNA-bound RNA samples over q-arm telomeric regions of chromosomes 2 and 10. The upper track of each panel shows the whole chromosome, and the tiny green bar to the right is the focused genomic region. The blue tracks show expression signals of three samples (normalized read frequencies per 10^7^ reads). The green track shows location of TERRA sequences in the mouse genome (minimum four consecutive TTAGGG repeats). The nucleotide sequence is also shown at the bottom of each panel with color bars (the grey bars show undetermined “N” nucleotides). (**b**) Heatmap showing the percent identity matrix between each pair of sequences of chromosomes12 and 10 (mice). (**c**) Sperm RNA-seq signals over TERRA sequences at p- and q-arm telomeric regions of several human chromosomes. Each track shows a different sperm sample (D_RNA_1 and 4 are DNA-bound RNAs, and RNA_1 and 4 are free cytoplasmic RNA samples). The heights of the peaks show normalized expression level (number of reads per 10^7^ reads) over each genomic region, with an equal 0–50 scale. The green track shows the location of TERRA telomeric repeats in the human genome (minimum four consecutive TTAGGG repeats). All human telomeric regions with a sequenced TERRA sequence in the human reference assembly GRCh38 are shown.

**Figure 5 cells-10-01556-f005:**
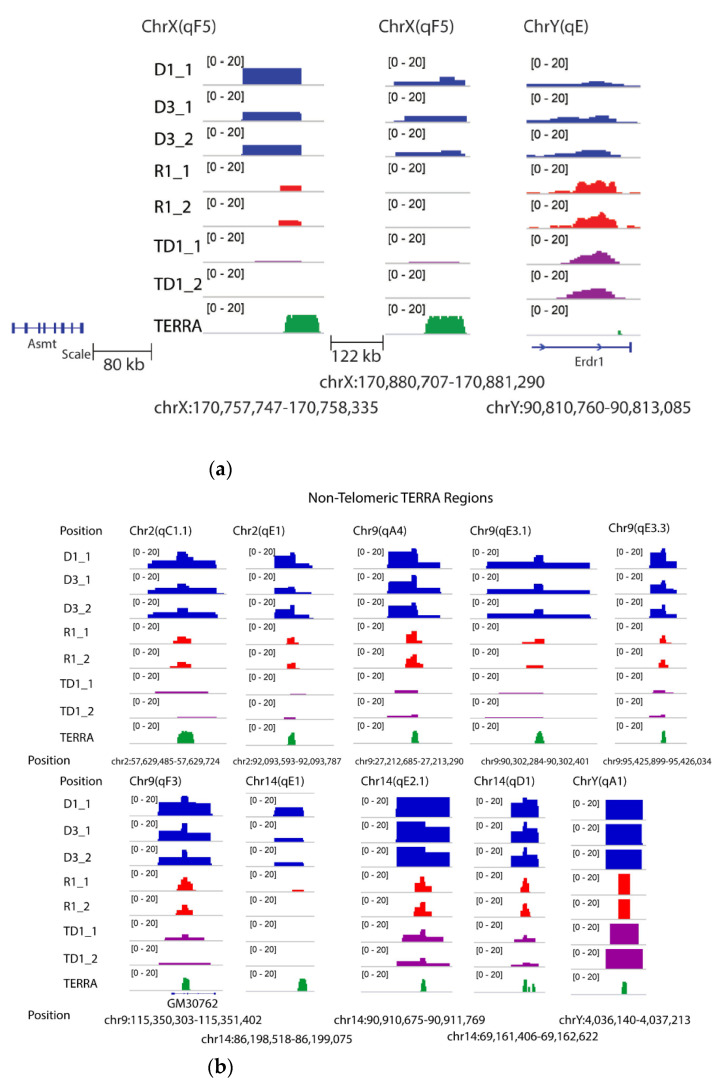
Non telomeric TERRA over mouse chromosomes. (**a**) TERRA spots are present in sex chromosomes. TERRA Regions on ChrX and ChrY. The number of TERRA foci co-localized with pseudoautosomal regions for instance, Asmt on X and Erdr1 on Y according to Lee et al [32]. (**b**) TERRA spots are present on chromosomes 2, 9, 14 and Y.

**Figure 6 cells-10-01556-f006:**
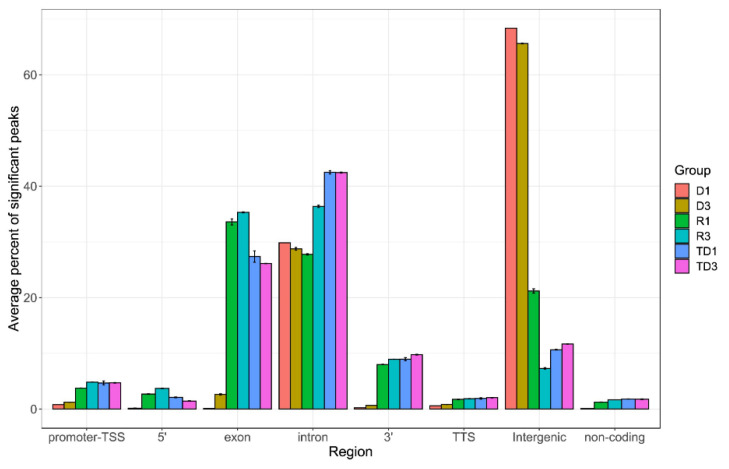
Peak annotation on entire genome. After obtaining peak sets, peak annotation was performed to associate with nearby genes and genomic features. A false discovery rate (FDR) cutoff of 0.001 was used for identifying significant peaks. Peak annotation was performed using Homer software and the annotate_peaks function to associate each peak with nearby genes and genomic features. Annotated positions for promoter-TSS, exons, introns, and other features were based on the mm10 transcripts. Enrichment analysis of associated genes was also performed using Enrichr for identifying Gene Ontologies (GOs) and pathways significantly over-represented by associated gene sets. For each sample group (D: DNA bound RNA in sperm; R: free RNA in sperm; TD: DNA bound RNA in testis), we measured the percent of significant peaks that were located in each genomic region. Y-axis shows the average percentage in biological replicates of each sample group. Error bars represent mean ± SD.

## Data Availability

https://www.ncbi.nlm.nih.gov/geo/query/acc.cgi?acc=GSE143919, will be released on 1 January 2023, accessed on 2 March 2021.

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
