# Peer review of "DNA-RNA Hybrid (R-Loop): From a Unified Picture of the Mammalian Telomere to the Genome-Wide Profile"

_cells, 2021, doi:10.3390/cells10061556_

Round 1
Reviewer 1 Report
This manuscript seems to present an exciting method to identify R-Loops throughout the genome. However, it is so poorly written that I am not sure. This manuscript needs to be rewritten and put together in a way that makes it much clearer or it will never be fully accepted or understood.
I think that the authors have presented a new method to detect R-loops. In lines 115-116, the authors describe the method very nicely, with one thing I do not yet understand. This method would be clearer if the authors could explain WHY only single stranded DNA or RNA binds to the nitrocellulose. It is crucial to the entire manuscript, and an explanation of why this is true would be helpful. As a control, it would be very convincing if the authors repeated the experiment in Fig. 1b, with one change, that they DID denature the DNA before transferring the DNA to the nitrocellulose. This should give a positive signal in all lanes, verifying the method does only identify R-loops. Because the entire manuscript is built on this method, I think this control is crucial to the manuscript.
As for the rest of the manuscript, I have several questions that could probably be resolved by much clearer writing.
- TERRA was never fully defined, other than to say “complementary RNA” in the Abstract (line 11).How did the authors come up this acronym? Is it only in telomeres?
- The data in Fig. 5 are very confusing.There are several different Fig. 5c’s, these should be better labeled. It is not clear (to me) where on the chromosomes these maps refer. Are the “non-telomeric regions” better described as “sub-telomeric regions” or are they truly throughout the chromosome.
- It also appears that all TERRA sites include the sequence UUAGGG repeat.How many repeats are necessary for this method to pick it up? The maps must be identified by nearby non-repetitive elements – is this true?
- There are two SFig. 1s.I assume they want to use the first? The method is not clear.
Author Response
This manuscript seems to present an exciting method to identify R-Loops throughout the genome. However, it is so poorly written that I am not sure. This manuscript needs to be rewritten and put together in a way that makes it much clearer or it will never be fully accepted or understood.
Thanks for reviewing our manuscript. We are sorry that our manuscript introduce trouble in reviewing process. We have tried to clarify important points raised by the reviewers but if still some sentences or paragraphs are problematic we will appreciate your help to improve.
I think that the authors have presented a new method to detect R-loops.
Yes, it is, you are right a simple method to reveal the RNAs contained in stable R-loops.
In lines 115-116, the authors describe the method very nicely,
Thank you,
with one thing I do not yet understand.
This method would be clearer if the authors could explain WHY only single stranded DNA or RNA binds to the nitrocellulose. It is crucial to the entire manuscript, and an explanation of why this is true would be helpful.
We have added sentences to explain in main text.
This is known for several decades see below history:
Southern blotting was introduced by Edwin Southern in 1975 as a method to detect specific sequences of DNA in DNA samples. The other blotting techniques emerged from this method have been termed as Northern (for RNA), Western (for proteins), Eastern (for post-translational protein modifications) and Southwestern (for DNA-protein interactions) blotting.
If alkaline transfer methods are used, the DNA gel is placed into an alkaline solution (typically containing sodium hydroxide) to denature the double-stranded DNA. The denaturation in an alkaline environment may improve binding of the negatively charged thymine residues of DNA to a positively charged amino groups of membrane, separating it into single DNA strands for later hybridization to the probe (see below), and destroys any residual RNA that may still be present in the DNA. The choice of alkaline over neutral transfer methods, see standard protocol online from sigma:
https://www.sigmaaldrich.com/technical-documents/articles/biology/southern-and-northern-blotting.html
In summary double stranded DNA is either purely or not really retained (especially large fragments of DNA) on the nitrocellulose membrane, while single-stranded DNA and RNA are negatively charged and will be retained to positive charge of the nitrocellulose membrane.
As a control, it would be very convincing if the authors repeated the experiment in Fig. 1b, with one change, that they DID denature the DNA before transferring the DNA to the nitrocellulose. This should give a positive signal in all lanes, verifying the method does only identify R-loops.
Yes, there is already this controls in Figure 1b at right. We have analysed samples on agarose gel since the NaOH treatment deforms the acrylamide gel.
Because the entire manuscript is built on this method, I think this control is crucial to the manuscript.
After electrophoresis and the agarose gel denaturation step, single stranded DNA will be attached to the nitrocellulose membrane. RNA molecules are degraded under alkaline condition.
As for the rest of the manuscript, I have several questions that could probably be resolved by much clearer writing.
We have modified text upon your suggestion.
- TERRA was never fully defined, other than to say “complementary RNA” in the Abstract (line 11).How did the authors come up this acronym? Is it only in telomeres?
Sorry, to introduce confusion and not clearly stated, TERRA is known as lncRNAs for Telomeric repeat-containing RNA (TERRA) is transcribed, at chromosome ends telomeres which consist of arrays of G-rich repeats (TTAGGG)n in a conserved manner from yeast to humans (Azzalin et al., 2007; Feuerhahn et al., 2010; Luke et al., 2008). TERRA is an RNA polymerase II (RNAPII) transcript that harbors sequence elements from the subtelomere and telomeric repeats (Luke et al., 2008),.
Although the biological function remains unknown, noncoding RNAs have also been reported at the telomeric
ends of several organisms, including birds (Solovei et al. 1994), trypanosomes (Rudenko and Van Der
Ploeg 1989), and mammals (Azzalin et al. 2007; Schoeftner and Blasco 2008). Interestingly, one group
reported that telomeric RNAs are enriched near the inactive X inmammals (Schoeftner and Blasco 2008).
- The data in Fig. 5 are very confusing. There are several different Fig. 5c’s, these should be better labeled. It is not clear (to me) where on the chromosomes these maps refer. Are the “non-telomeric regions” better described as “sub-telomeric regions” or are they truly throughout the chromosome.
We have proposed the new figure for figures 5.c indicating several non-telomeric TERRA repeats distributed throughout the genome on chromosomes 2, 9, 14 and Y. They are not located in the sub-telomeric regions. The position on the chromosomes is marked at the bottom of each location.
- It also appears that all TERRA sites include the sequence UUAGGG repeat.
Yes, correct, all TERRA regions contain minimum 4 consecutive TTAGGG repeats.
- How many repeats are necessary for this method to pick it up? The maps must be identified by nearby non-repetitive elements – is this true?
Minimum 24mer versions of TERRA containing the minimal 4 repeats (TAACCC) necessary to form respective G-quadruplex telomere structure (H Martadinata · 2013, Structure of Human Telomeric RNA (TERRA): Stacking of Two G-Quadruplex Blocks in K+ Solution)
In our study, the genome-wide locations of TERRA repeats were profiled by aligning 24 and 48 bp length sequences composed of 4 and 8 copies of TERRA on the reference genome using bowtie2 using argument “-a” to signal any alignment in the genomes.
- There are two S Fig. 1s.I assume they want to use the first? The method is not clear.
Thanks for the note, we have improved the caption, the first now Supplementary Figure 1a is for molecular biology analysis with DNA and RNAs molecules and the second Figure 1b is for RNAs preparation for RNAseq analysis.
Reviewer 2 Report
They develop a method for profiling R-loop complexes whithout the need of antibodies or RNaseH treatment avoiding the problem of these techniques. This method has provided useful for profiling the telomeric TERRA complex.
The method has proven to be valuable for this kind of studies and deserves publication.
I do not know if the arrangement of figures on the manuscript is the final form, but in the actual arrangement there is a waste of space.
There is also disproportionate scaling between subfigures, for instance fig. 1d can be scaled down so that the band size are similar to those in the other figures.
Maybe most of the figure 5, or the complete figure, can be supplementary material.
Author Response
They develop a method for profiling R-loop complexes whithout the need of antibodies or RNaseH treatment avoiding the problem of these techniques. This method has provided useful for profiling the telomeric TERRA complex.
The method has proven to be valuable for this kind of studies and deserves publication.
I do not know if the arrangement of figures on the manuscript is the final form, but in the actual arrangement there is a waste of space.
Thanks for reviewing our manuscript, we propose a new version for the figures.
There is also disproportionate scaling between subfigures, for instance fig. 1d can be scaled down so that the band size are similar to those in the other figures.
Right, we agree, done in new version.
Maybe most of the figure 5, or the complete figure, can be supplementary material.
We propose a new composition for figure 5.
Please see the attachment

Round 2
Reviewer 1 Report
The authors have addressed all my concerns and I would suggest the manuscript be accepted.